# Tolerance of Infants Fed a Hydrolyzed Rice Infant Formula with 2′-Fucosyllactose (2′-FL) Human Milk Oligosaccharide (HMO)

**DOI:** 10.3390/nu16121863

**Published:** 2024-06-13

**Authors:** Carlett Ramirez-Farias, Jeffery S. Oliver, Jane Schlezinger, John T. Stutts

**Affiliations:** Scientific & Medical Affairs, Abbott Nutrition, Columbus, OH 43219, USA; jeffery.oliver@abbott.com (J.S.O.); john.stutts1@abbott.com (J.T.S.)

**Keywords:** cow’s milk allergy, hydrolyzed rice formula, human milk oligosaccharides, HMO, 2′-FL

## Abstract

Background: The purpose of this research was to assess the growth, tolerance, and compliance outcomes associated with the consumption of a hydrolyzed rice infant formula (HRF) enriched with 2′-Fucosyllactose (2′-FL) a Human Milk Oligosaccharide (HMO), and nucleotides in an intended population of infants. Methods: This was a non-randomized single-group, multicenter study. The study formula was a hypoallergenic HRF with 2′-FL, Docosahexaenoic acid (DHA), Arachidonic acid (ARA), and nucleotides. Infants 0–90 days of age who were formula fed and experiencing persistent feeding intolerance symptoms, symptoms of suspected food protein (milk and/or soy) allergy, or other conditions where an extensively hydrolyzed infant formula was deemed an appropriate feeding option were recruited by pediatricians from their local populations. The primary outcome was maintenance of weight-for-age z-score. Weight, length, head circumference, formula intake, tolerance measures, clinical symptoms and questionnaires were collected. Thirty-three infants were enrolled, and 27 completed the study, on study product. Results: Weight-for-age z-scores of infants showed a statistically significant improvement from Visit 1 to Visit 4 (*p* = 0.0331). There was an adequate daily volume intake of 762 ± 28 mL/day, average daily number of stools of 2.1 ± 0.3, and mean rank stool consistency of 2.38 ± 0.18. After 28 days of switching to a HRF, 86.8 ± 5.9% of the symptoms resolved or got better by Visit 4 as reported by parents. Conclusions: HRF with 2′-FL HMO was safe, well tolerated, and supported weight gain in infants with suspected cow’s milk allergy or persistent feeding intolerance.

## 1. Introduction

Cow’s milk allergy (CMA) is one of the most common food allergies in infants, and the prevalence ranges from approximately 1–7.5% [1]. If not appropriately managed, it can adversely impact growth and development. Recognition of the potential benefits of a hydrolyzed rice protein-based formula (HRF) for infants with CMA began with the insight that rice is a protein with low allergenicity [2,3]. Preclinical research with hydrolyzed rice-protein [4] paved the way for several clinical studies that evaluated HRF as a plant-based alternative, free from cow’s milk protein for CMA. The use of hydrolyzed rice formulas has been endorsed by experts in the field [1,2,3] and in the DRACMA [1,5,6,7] and ESPGHAN [8] guidelines. HRFs have been used in Europe since the 2000s as a nutritional equivalent to cow’s milk protein-based extensively hydrolyzed formula (eHF) and for the dietary management of CMA. All infant formula used in the management of CMA must comply with country specific regulations and be nutritionally complete [2].

Human milk oligosaccharides (HMOs) are the third most abundant component in human milk after lactose and lipids. They are present at about a 20-fold higher concentration in human milk than in bovine milk [9,10]. HMOs are non-digestible carbohydrates and are substrates for intestinal microbiota, such as Bifidobacterium and Bacteroides [10,11]. Mature human milk contains 12–15 g/L of HMOs of which 2′-fucosylactose (2’-FL) is the most abundant ranging from 0.06 to 4.65 g/L [9,12,13]. Levels of HMOs vary over time in human milk [9,11,13], and between mothers [14]. The different phases of lactation and maternal phenotypic secretor types play important roles in HMO concentration and profile [15].

In previous growth and tolerance clinical studies, infants fed 2′-FL fortified formulas showed no significant differences in weight, length and head circumference, between the experimental formula (added 2′-FL) and control formula [16,17]. Other studies in healthy infants fed infant formula with added 5 HMOs have also demonstrated normal growth parameters [18]. A clinical study conducted in infants with suspected cow’s milk allergy who were fed an extensively hydrolyzed formula with 2′-FL demonstrated adequate growth and tolerance [19]. Also, no safety concerns were observed. Healthy infants fed a formula containing 2′-FL had circulating cytokine concentrations which differed from the control fed infants but did not differ from the breast-fed infants [20]. Preclinical studies suggest that HMOs may directly interact with the immune system [21,22,23]. Human milk oligosaccharides, such as 2′-FL and 6′-SL may reduce the symptoms of allergy through induction of IL-10 regulatory cells and/or indirect stabilization of mast cells in pre-clinical models [23]. 

The purpose of this study was to evaluate the growth, tolerance, compliance, and change in clinical symptoms in infants with suspected food protein allergy or persistent feeding intolerance fed a HRF powdered infant formula with 2′-FL, docosahexaenoic acid (DHA), arachidonic acid (ARA), and nucleotides.

## 2. Materials and Methods

Between June 2022 and May 2023, 33 infants were enrolled in a single group, non-randomized, multicenter center study with a 28-day feeding intervention. The protocol was conducted in accordance with Good Clinical Practices.

This study was approved by the Institutional Review Board (IRB) and was registered at clinicaltrials.gov (accessed on 11 May 2022) (NCT05369494). All participants’ parents gave written informed consent approved by an Independent Ethics Committee/IRB and signed Health Insurance Portability and Accountability Act (HIPAA) documents.

### 2.1. Study Population

Infants were recruited by the investigators from their local populations.

Inclusion criteria included formula-fed infants less than 90 days of age who were either Group A) experiencing persistent feeding intolerance symptoms, symptoms of suspected food protein (milk and/or soy) allergy or other conditions where an extensively hydrolyzed formula (EHF) was deemed appropriate by the healthcare professional or Group B) already consuming an EHF (Alimentum^®^, Nutramigen^®^, or another commercially available EHF on the market) for persistent feeding intolerance symptoms, symptoms of suspected food protein (milk and/or soy) allergy, or other conditions where EHF was deemed an appropriate feeding option. 

Parent(s) of infants confirmed their intention not to administer prescription medications, over-the counter (OTC) medications, home remedies (such as juice for constipation), prebiotics, probiotics, herbal preparations or rehydration fluids that might affect gastrointestinal (GI) tolerance, unless the infants were currently consuming and had been directed by their health care professional to continue their use during the study. Parent(s) also confirmed their intention not to administer solid foods, juices, vitamins, and minerals supplements (except for vitamin D). Infants received the study formula ad libitum as their sole source of nutrition.

Infants were excluded if there was an adverse maternal, fetal or participant medical history thought by the investigator to have potential effects on growth, and/or development. Exclusion criteria included awareness of a positive drug screen in the mother of the participant, suspected maternal substance abuse including alcohol, if the infant was receiving oral or inhaled steroids, if they had received an amino acid-based formula, or if the infant had an allergy or intolerance to any ingredient in the study product, as reported by the parent.

### 2.2. Methods

The study product was a clinically labelled hydrolyzed rice powder infant formula designed to provide 20 kcal per fl oz at standard dilution with 0.2 g/L of 2′-FL, DHA, ARA, and nucleotides (Similac^®^ Arize^®^, Abbott Nutrition, Abbott Laboratories, Columbus, OH, USA). This study enrolled subjects from 6 outpatient clinical sites in the United States. This study formula is available currently outside of the United States.

Weight and length measurements were performed by trained personnel and plotted on growth curves using the 2006 World Health Organization (WHO) reference data [24].

The primary outcome of this study was maintenance of weight-for-age z-score throughout the study. The secondary variables were mean rank stool consistency (MRSC), predominant stool color, average number of stools per day, percent of feedings with spit-up/vomit associated with feeding (within 1 h), weight, interval weight gain per day, length, and interval length gain per day. WHO z-scores were calculated for weight-for-age, length-for-age, and weight-for-length.

Supportive and demographic variables were the following: average volume of study formula intake per day (mL/day and mL/kg/day), as well as parental responses to the Infant Feeding and Stool Patterns Questionnaire, the Formula Satisfaction Questionnaire and Clinical Symptoms Questionnaire. Infant Feeding and Stool Patterns Questionnaires were used with a “Likert Scale”. Parents ranked each question with the following alternatives “(1) Always, (2) Frequently, (3) Some of the time, (4) Rarely, (5) Never”. Clinical symptoms reported by the parents were reviewed at enrollment, at Visit 3 (after approximately 14 days on study formula) and Visit 4 (after approximately 28 days on study formula) by the study investigators. Subject demographic and past medical history information (age, weight, and length at enrollment, birth weight, birth length, gestational age, mode of delivery, pre-study feeding history including number of days on each feeding, gender, race, and ethnicity) were collected (see Table 1) Safety monitoring consisted of the collection of adverse events and serious adverse events during the study.

### 2.3. Statistical Methods

A total sample size of 15 participants had 80% power to detect a mean change of −0.303 in weight-for-age z-scores from study entry to exit assuming that the common standard deviation is 0.39 (CP-AK94), using a two-sided 5% level paired *t*-test. Based on an estimated 50% attrition rate, enrollment of 30 infants was planned. The nQuery Advisor^®^ 8 software was used in these determinations. All study results presented will correspond to the protocol evaluable (PE) cohort.

## 3. Results

There were 33 infants who enrolled in the study. 

Weight-for-age z-scores of infants showed a statistically significant improvement from Visit 1 to Visit 4 (Mean change ± standard error, SEM: 0.35 ± 0.15, *p* = 0.0331 from the evaluable analysis). In addition, 89% of the infants had a change in weight-for-age z-score greater than −0.303. Furthermore, since the result of the comparison with 0 is statistically significant, this means that the comparison with −0.303 is also statistically significant (the evolution of weight-for-age z-score throughout the study is included in Figure 1 and Table 2). All variables are displayed, unless otherwise stated, to show the data from the visit when both groups (A + B) initiated study product, until Visit 3 and Visit 4. For Group A, the initiation of study product was at Visit 1, and for Group B, it was at Visit 2. Group B participants were already on an extensively hydrolyzed formula when recruited for the study. Hence, they were started on the study formula at Visit 2, allowing three days to collect prospective data on their eHF formula intake before switching to study formula. The intent was not to compare Group A and Group B, but to assess how these infants responded to the experimental formula collectively. Both groups had approximately 28 days of study formula feeding.

From the above table, we show average anthropometric, intake and stool data. Average MRSC was based on parental reported daily stool records, starting from when the infants initiated study product (Visit 1 for Group A and Visit 2 for Group B) to Visit 4 (period spans approximately 28 study feeding days). 

From initiation of study product to Visit 4, there was a mean of 31.7 ± 7.1 percent of study formula feedings per day that had associated spit-up or vomit. To put into context, there are several infant clinical studies that have recorded the frequency of spit-up/vomit as percent of feeding, and the percent ranges from 16% to 85% in healthy infants [25,26]. The study that reports 85% identified their subjects as healthy normal infants who were heavy spitters [26]. Another published study [27] evaluated a partially hydrolyzed rice protein-based formula versus an intact cow’s milk protein-based formula in healthy infants and reported a frequency of spit-up/vomit as a percent of feeding ranging between 16.3 and 37.6%. Spit-up/vomit is also common among breastfed infants and some publications have reported percents between 10% and 60% [16,28].

The weight-for-age z-score was −0.62 ± 0.20 at Visit 1, −0.50 ± 0.17 at Visit 3 and −0.26 ± 0.21 at Visit 4. The length-for-age z-score was −0.62 ± 0.29 at Visit 1, −0.58 ± 0.25 at Visit 3, and −0.08 ± 0.30 at Visit 4 (See Table 2). The average weight gain per day from Visit 1 to Visit 4 was 35.0 ± 3.4 g per day, and the length gain per day from Visit 1 to Visit 4 was 0.13 ± 0.01 cm per day. The mean ± SEM number of study formula feedings per day was 6.6 ± 0.3 from initiation of study product to Visit 3 and 6.5 ± 0.3 from initiation of study product to Visit 4. The average volume of study formula intake per day was 713 ± 29 mL/day from initiation of study product to Visit 3 and 762 ± 28 mL/day from initiation of study product to Visit 4. The adjusted average volume of study formula from Visit 1 to Visit 4 was 150.3 ± 7.3 mL/kg/day (See Table 3). This volume of study formula is as expected, as a fluid intake of 150 mL/kg/day in infancy should be maintained to achieve optimum nutrition. The opinion of the EFSA panel on Dietetic Products, Nutrition and Allergies have set Adequate Intakes (AIs) for infants in the first year of life and estimate this to be 100–190 mL/kg/day [29]. 

Clinical history and symptoms were recorded at Visit 1, Visit 3, and Visit 4. It is important to note that at Visit 4, compared to baseline, parents reported 100% of infants with clinical symptoms as better or resolved for rash/eczema, constipation, diarrhea, blood in stool and vomiting. Spit-up/gagging/retching resolved or improved in 85% of infants and remained the same in 15% of infants. Fussiness improved or resolved in 94% of infants and remained the same in 6% of infants (see Table 4). 

After 28 days of switching to the HRF, 86.8 ± 5.9% of the symptoms resolved or got better by Visit 4 (see Table 5). Overall, at Visit 4 (28 days of study formula) all persisting symptoms either remained the same, improved or resolved, and none of the symptoms worsened. Regarding how well the infant did on the formula, the Formula Satisfaction Questionnaire showed parental responses of ‘well’ and ‘very well’, improving from 16% at enrollment to 74% at the end of the study. Of note, stress levels were also assessed by a Parental Perspective Questionnaire. Forty-two percent of parents assessed their stress level as high at enrollment vs. 10% at Visit 4.

## 4. Discussion

To the knowledge of the authors, this is the first clinical study conducted to date with a HRF with 2′-FL HMO (study product). 

The primary outcome of this study was achieved (maintenance of weight-for-age z-score, from study initiation (Visit 1) to the end of the study period (Visit 4)). Not only did the infants maintain their weight-for-age z-scores, but they also showed a statistically significant improvement from Visit 1 to Visit 4 (*p* = 0.0331). This is an important finding. Infants with cow’s milk protein allergy (CMA) are at risk of impaired growth [30,31] and a study by Meyer et al. found that 11% of children with food allergies are stunted [32]. In another study, DuPont [2] highlights that growth may be affected during the elimination diet. It is reassuring that our data demonstrated an average weight gain per day of 35.0 ± 3.4 g and an average length gain per day of 0.13 ± 0.01 cm, both within the WHO 2006 recommendations for optimum growth for infants 0–24 months of age [24]. Other studies have shown similar findings in terms of adequate growth in both healthy infants [2,27] as well as infants with CMA [33,34,35,36,37,38,39] while being fed a HRF. More recently, Nieto-Garcia et al. conducted a study on the use of a hydrolyzed rice formula over a 12-month period. A double-blind randomized controlled trial with a 12-month follow-up in cow’s milk allergic infants comparing a HRF with an eHF which showed no significant differences in growth parameters [39].

HRF has been safely used in European countries such as Italy and Spain for the past 20 years. Due to the historical lack of availability of HRF in many other countries, the dietary management of infants with CMA has largely been dominated by extensively hydrolyzed infant formula (eHF) containing cow’s milk peptides, or amino acid-based formula (AAF). However, recently published infant guidelines [6] for the management of CMA acknowledge that some infants with a proven diagnosis of CMA have incomplete resolution of symptoms upon management with a particular eHF and note that data from the UK report a 29% failure rate of some eHFs. They go on to recommend, along with other guidelines [1,40,41] that as HRFs have become more widely available, they are now an alternative feeding option for the treatment of CMA. The study formula is available currently in some countries outside of the United States.

In this study, infants in group B were fed and eHF for 17.9 ± 3.8 days before switching to a HRF with 2′-FL HMO (study product) at the beginning of the study period. At Visit 4, group B had a weight-for age z-score of −0.19 ± 0.33 vs. −0.30 ± 0.28 in group A, and length-for-age z-scores in group B were –0.24 ± 0.42 vs. 0.19 ± 0.43 in group A. These results reinforce that early nutritional intervention in CMA with a hypoallergenic formula can lead to improvement in growth parameters. As per guidelines, [1,6,40] which recommend the use of an eHF as part of a 2–4 week elimination diet for suspected CMA, group B in this study followed this recommendation but remained symptomatic. Some International guidelines continue to recommend that in cases of failure of an eHF, an AAF may be indicated. In this clinical study, infants in group B, did not achieve symptom resolution with an eHF, were switched to a HRF with 2′-FL HMO (study product) and achieved clinical symptom improvement, without the need to switch to an AAF. This suggests that HRF with 2′-FL HMO is an appropriate alternative to eHF for infants with persistent clinical symptoms when an eHF has not let to symptom resolution. Importantly, HRFs are devoid of cow’s milk protein, like AAF. The protein source in HRF is derived from hydrolyzed rice. Therefore, an additional advantage of using a HRF for infants with CMA, is that allergic reactions to cow’s milk protein would be unlikely. 

Feeding difficulties are a common parental complaint in food allergic children, particularly in those with CMA under elimination diets [32]. In fact, one study [42] showed a significant difference in allergic children with altered taste preferences compared to non-allergic children. Infant refusal of hypoallergenic formula, due to the bitter taste of some hydrolysates, can be a common problem among infants that can impact formula intake and consequently, growth and development. Our study findings demonstrated that the HRF used in this study did not impact formula intake. The average number of study feedings per day and total volume intake of study formula per day were within the expected parameters to promote growth as per infant feeding recommendations [43]. This supports that the study formula was well tolerated and accepted by the infants. 

Tolerance of the study formula was reported by asking the parents to register their infants’ symptom improvement and/or resolution once their infant(s) were switched to the study formula. Fussiness, vomiting, eczema or rash, constipation, diarrhea, and blood in stool all improved or resolved by the end of the study period. These reassuring parental reports provide further support for the efficacy of HRF in achieving improvements and/or resolution in the above listed clinical symptoms. 

Another important finding of the study was the reduction in parental reporting stress levels, where 42% of parents assessed their stress level as high at enrollment vs. 10% at the end of the study period. Many studies [44,45] have shown that living with a food allergy significantly impacts psychological (as well as emotional and physical) aspects of quality of life (QoL). Food allergy is associated with a decreased caregiver QoL and milk allergy has been associated with worse total specific QoL scores vs. peanut allergy [46].

We acknowledge there were limitations to this study. Firstly, it was a one arm study. Secondly, the time the infants were on the study product was of short duration. While one month is enough to assess tolerance, as well as efficacy of a hydrolyzed formula as per current guidelines [1,6,40] it is not enough time to assess long term growth, which was not the aim of this study. However, this study did demonstrate improvements in short-term growth. Further studies are needed to assess the potential benefits of 2′-FL HMO on different areas, such as, changes in the microbiome and the immune system by the modulation of cytokines in randomized control trials.

## 5. Conclusions

To the knowledge of the authors, this is the first clinical study conducted to date with a HRF with 2′-FL HMO. This study shows that a HRF with added 2′-FL, DHA, ARA and nucleotides was safe, well accepted and enabled a statistically significant improvement of weight-for-age z-scores in infants with suspected CMA and/or persistent feeding difficulties. Parental reports also indicated that the formula was well accepted by the infants. 

## Figures and Tables

**Figure 1 nutrients-16-01863-f001:**
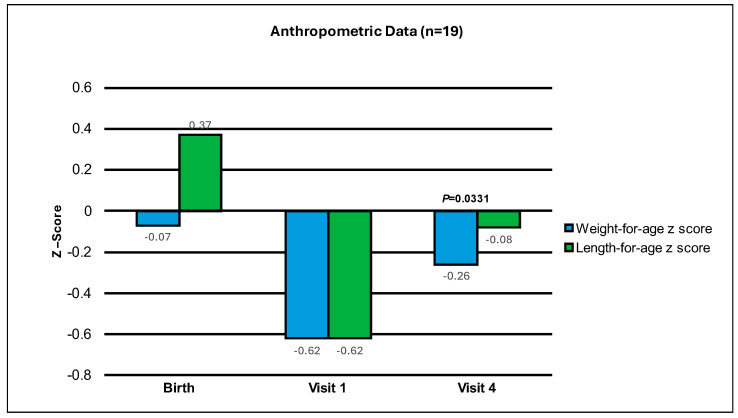
Mean change in weight-for-age and length-for-age z-scores from Birth, to Visit 1 and Visit 4 (end of study period).

**Table 1 nutrients-16-01863-t001:** Demographic Data.

	Group A (No Prior EHF) (N = 12)	Group B (Prior EHF) (N = 7)	Total (N = 19)
Sex, n (%)			
Male	7 (58.3)	5 (71.4)	12 (63.2)
Female	5 (41.7)	2 (28.6)	7 (36.8)
Ethnicity, n (%)			
Hispanic/Latino	4 (33.3)	0 (0.0)	4 (21.1)
Not Hispanic/Latino	8 (66.7)	6 (85.7)	14 (73.7)
Not reported	0 (0.0)	1 (14.3)	1 (5.3)
Race, n (%)			
Black/African American	2 (16.7)	3 (42.9)	5 (26.3)
White	9 (75.0)	4 (57.1)	13 (68.4)
Black/African American/White	1 (8.3)	0 (0.0)	1 (5.3)
Mode of Delivery, n (%)			
Vaginal	10 (83.3)	5 (71.4)	15 (78.9)
C-section	2 (16.7)	2 (28.6)	4 (21.1)
Gestational Age: Weeks			
Mean ± SEM	38.7 ± 0.4	37.0 ± 0.6	38.1 ± 0.4
Min, Max	37.0, 40.0	35.0, 40.0	35.0, 40.0
(95% CI)	(37.9, 39.4)	(35.6, 38.4)	(37.3, 38.8)
Age at enrollment (d)			
Mean ± SEM	52.1 ± 5.5	44.7 ± 6.2	49.4 ± 4.1
Min, Max	16.0, 80.0	29.0, 75.0	16.0, 80.0
(95% CI)	(39.9, 64.2)	(29.6, 59.8)	(40.7, 58.1)
Birth Weight (g)			
Mean ± SEM	3360 ± 148	3195 ± 200	3299 ± 117
Min, Max	2438, 4139	2438, 3884	2438, 4139
(95% CI)	(3035, 3684)	(2704, 3685)	(3053, 3545)
Birth Length (cm)			
Mean ± SEM	50.2 ± 0.9	50.5 ± 0.9	50.3 ± 0.6
Min, Max	47.0, 55.9	47.0, 53.3	47.0, 55.9
(95% CI)	(48.3, 52.1)	(48.4, 52.7)	(49.0, 51.6)
Number of days on current feeding			
Mean ± SEM	39.0 ± 5.8	17.9 ± 3.8	31.2 ± 4.5
Min, Max	1.0, 69.0	1.0, 30.0	1.0, 69.0
(95% CI)	(26.3, 51.7)	(8.5, 27.2)	(21.7, 40.7)
Number of days on other feedings			
Mean ± SEM	18.8 ± 4.8	28.4 ± 8.1	22.3 ± 4.3
Min, Max	0.0, 51.0	12.0, 64.0	0.0, 64.0
(95% CI)	(8.1, 29.4)	(8.7, 48.2)	(13.3, 31.3)

SEM = standard error of the mean. CI = confidence interval. Min = minimum. Max = maximum.

**Table 2 nutrients-16-01863-t002:** Anthropometric weight-for-age and length-for-age z-score.

	Group A (No Prior EHF) (N = 12)	Group B (Prior EHF) (N = 7)	Total (N = 19)
Weight for age z-score, mean ± SEM			
Birth	0.07 ± 0.29	−0.31 ± 0.42	−0.07 ± 0.24
Visit 1	−0.66 ± 0.31	−0.54 ± 0.20	−0.62 ± 0.20
Visit 3	−0.54 ± 0.24	−0.44 ± 0.23	−0.19 ± 0.33
Visit 4	−0.30 ± 0.28	−0.19 ± 0.33	−0.26 ± 0.21
Length for age z-score, mean ± SEM			
Birth	0.32 ± 0.42	0.46 ± 0.44	0.37 ± 0.30
Visit 1	−0.58 ± 0.42	−0.69 ± 0.34	−0.62 ± 0.29
Visit 3	−0.51 ± 0.36	−0.69 ± 0.28	−0.58 ± 0.25
Visit 4	−0.24 ± 0.42	0.19 ± 0.43	−0.08 ± 0.30

**Table 3 nutrients-16-01863-t003:** Anthropometric, intake and stool data.

Group A + Group B Combined Data	Visit 1 to Visit 4 (N = 19)
Change in weight-for-age z-score	0.35 ± 0.15
Change in length-for-age z-score	0.54 ± 0.20
Weight gain per day (grams)	35.0 ± 3.4
Number of feedings per day on study formula	6.5 ± 0.3
Average volume intake of study formula (mL/day)	762 ± 28
Average volume intake of study formula (mL/kg/day)	150.3 ± 7.3
Average number of stools per day	2.1 ± 0.3
Mean rank stool consistency (MRSC)	2.38 ± 0.18

**Table 4 nutrients-16-01863-t004:** Change in Clinical Symptoms from Baseline to Visit 3 and Visit 4 (Group A + Group B).

Clinical Symptoms	At Study Initiation	At Visit 3 (n (%) of Subjects with the Symptom at Entry)	At Visit 4, n (%) of Subjects with the Symptom at Entry)
Diarrhea	7 (37%)	2 (29%)—Same 4 (57%)—Better 1 (14%)—Resolved	3 (43%)—Better 4 (57%)—Resolved
Constipation	10 (53%)	2 (20%)—Same 3 (30%)—Better 5 (50%)—Resolved	4 (40%)—Better 6 (60%)—Resolved
Blood in stool	1 (5%)	1 (100%)—Resolved	1 (100%)—Resolved
Vomiting	6 (32%)	2 (33%)—Better 4 (67%)—Resolved	1 (17%)—Better 5 (83%)—Resolved
Spit-up/Gagging/ Retching	13 (68%)	3 (23%)—Same 9 (69%)—Better 1 (8%)—Worse	2 (15%)—Same 9 (69%)—Better 2 (15%)—Resolved
Fussiness	16 (84%)	6 (38%)—Same 8 (50%)—Better 2 (13%)—Resolved	1 (6%)—Same 10 (63%)—Better 5 (31%)—Resolved
Rash or Eczema	4 (21%)	4 (100%)—Better	4 (100%)—Better
Other	1 (5%)	1 (100%)—Better	1 (100%)—Better

**Table 5 nutrients-16-01863-t005:** Clinical Symptoms: Percentage of Symptoms Resolved or Better (Group A + Group B).

% of Symptoms Resolved or Better at Visit 4 since Visit 1	
Mean ± SEM	86.8 ± 5.9
Median	100.0
Q1, Q3	75.0, 100.0

## Data Availability

The data that support the findings of this study are available upon request from the corresponding author (C.R.-F).

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
