# Peer review of "Tolerance of Infants Fed a Hydrolyzed Rice Infant Formula with 2′-Fucosyllactose (2′-FL) Human Milk Oligosaccharide (HMO)"

_nutrients, 2024, doi:10.3390/nu16121863_

Round 1

Reviewer 1 Report

Comments and Suggestions for Authors

Thank you for the article. As you have mentioned in your introduction, it is unlikely that hydrolyzed rice formula or a formula containing 2'FL, DHA, ARA and nucleotides will be a problem as there are already many publications on such formulations. I agree that though there is no current publication on a hydrolyzed rice formula containing a HMO, DHA, ARA and nucleotides, given what has been published, such a formulation is likely to be able to support growth. As such, the study design should probably not only address whether it will be well tolerated, support normal growth but also be effective as a substitute to current recommendations given for cow milk allergy. As such, the study should probably go on for a longer period then 28 days and should have a control group.

You have stated that your aim was to evaluate growth, tolerance, compliance  and change in clinical symptoms in infants suspected of food protein allergy, presumably cow's milk, and persistent feeding intolerance. A 28 day study period is rather short to evaluate growth adequately. For a study that was trying to evaluate compliance to the formula, there was a rather high drop out rate or patients failing evaluability criteria at 42%. Also a rather high number of infants appear to have feeding issues with the formula at around 32%. 

The methodology was not clear. There is supposed to be 4 visits to the clinic, presumably Visit 1 was at enrolment, Visit 3 was about 14 days and Visit 4 was at 28 days. It is not clear when Visit 2 takes place. The participants were also divided into 2 groups, A and B, based on whether prior extensively hydrolyzed formula had been given. The infants in Group B were only started on the study formula at Visit 2 and it is not clear why this was so. Furthermore, the numbers enrolled into the study are very small which makes it difficult for you to make any good conclusions. You had to analyse the clinical symptoms combined as the numbers were too small. If you do see an improvement in symptoms, can you conclude whether it is the hydrolyzed rice formulation or the addition of HMO or both that is contributing to the improvement? Is this formula really better than the extensively hydrolyzed formula that the infants were previously on in Group B (you will need to analyse this group separately to answer this question).

Reviewer 2 Report

Comments and Suggestions for Authors

The manuscript written by Aleman et al Tolerance of Infants Fed a Hydrolyzed Rice Infant Formula 2
with 2’-Fucosyllactose (2'-FL) Human Milk Oligosaccharide 
is dealing with an important health issue. The article is well organized, the experimental methods are suited and the conclusions are supported by the research data.

However, there are a few things that need improvement:

  1. Statistical analysis- authors state that the t-test was applied between the groups, however, it is not specific if the t-test was dependent and independent.
  2. Check the abbreviations throughout the manuscript and introduce the abbreviation when the full word appears the first time.
  3. The introduction part appears less informative about rice based ingredients on infant fomulas, thus this section should be indicated as detailed to understand the manuscript clearly. In the introduction, the authors should cite recent prevalence or incidence data.

Abstract

There is no background of the study; no clear objectives; nothing on review findings; no conclusions; no recommendation. As such, this abstract needs improvement.

Keywords can be improved.

Introduction

There is no section highlighting the need of performing this review.

No appropriate background; relevant studies are not cited.

Authors provided a very general introduction for each section; in-depth discussion is missing. All other sections are more general and covered many general aspects instead of focusing on objectives of the review.

In the last paragraph of the introduction which should define the aim of the study design. Then, this aim is forgotten throughout the manuscript.

Conclusion

Suggestion to regroup, restructure in a logical, natural sequence to become easy to read and understand.

Reviewer 3 Report

Comments and Suggestions for Authors

The manuscript provides valuable insights into the efficacy and safety of a novel hydrolyzed rice infant formula (HRF) enriched with 2'-FL human milk oligosaccharide (HMO) for infants with suspected cow's milk allergy (CMA). The study also has some limitations that should be taken into account, e.g. the study duration of 28 days may not be sufficient to fully assess long-term growth and development outcomes. Furthermore, the nonrandomized single-group design presents the possibility of selection bias and limits the ability to draw causal conclusions. Further research is needed to confirm these results and examine the long-term effects of this novel formula.

·         Introduction – While mentioning CMA in children is technically correct, the title focuses on infants. In the first paragraph, emphasize CMA in infants. Briefly mention the limitations of the current HRF formulas. Mention how 2'-FL could complement HRF in the treatment of CMA, highlight the potential benefits of 2'-FL, particularly for infants with CMA, and mention the reasons for including 2'-FL in HRF formulas for infants with CMA. Mention the structural variations of 2'-FL (e.g. 2'FL and 3-FL) to get a more complete picture. You could add a sentence about the possible mechanisms by which 2'-FL might improve tolerance. Consider adding a closing sentence that flows seamlessly into the statement of purpose.

·         Materials and Methods – Consider specifying recruitment locations (e.g., hospitals, clinics). Briefly mention the reasons for including infants already receiving EHF. Consider shortening the list of restricted medications/supplements by mentioning categories and referring to a separate document for details. Indicate which WHO growth curves were used. Consider reporting whether anthropometric measurements were performed by trained personnel. Briefly explain the scoring system used for the Infant Feeding and Bowel Behavior Questionnaire.

·         Results – Consider simplifying the statement in lines 153-158 for statistical significance and focus on the main outcome (improvement in Z-score by weight and age). On lines 175-183, consider condensing the average MRSC, number of bowel movements, and prevailing consistency information into a single table for readability. In lines 193-194 explain the reasons for including the adjusted average volume of food intake per kg/day. Consider combining this data in lines 220-223 with Table 5 to resolve/improve symptoms. Consider adding a short introductory sentence that summarizes the overall findings.

·         Discussion – Consider shortening this section for clarity. Briefly mention the WHO reference for the growth parameters used (e.g. weight for age and length for age). Mention the limitations of using WHO growth standards. Explain the statistical significance of Group B's Z-scores being closer to 0. Mention the mechanism behind altered taste preferences in allergic children (e.g. possible role of inflammation). Conclude the discussion by reviewing the key findings and potential role of HRF with 2'-FL for CMA management. Briefly discuss any limitations of the study and mention possible future research directions.

·         Conclusions – First, emphasize the novelty of the study and state that it is the first clinical trial to evaluate a hydrolyzed rice infant formula (HRF) with additional 2'-fucosyllactose (2'-FL) human milk oligosaccharide (HMO) evaluated in infants with suspected feeding protein allergy or persistent feed intolerance. Summarize the main objective of the study and key findings in a single, descriptive sentence. Discuss the clinical implications of the results and highlight the potential benefits of HRF with 2'-FL-HMO for infants with suspected food protein allergy or persistent feed intolerance. Briefly acknowledge any limitations of the study. Suggest potential areas for future research. Suggest investigations into the mechanisms by which HRF with 2'-FL-HMO may improve growth, tolerance, and clinical symptoms.

·         Abstract – The summary currently emphasizes growth, but could be improved by mentioning specific outcomes related to tolerance and compliance (e.g., symptom improvement, nutritional intake, parental satisfaction). Mention the observed improvements in tolerance and compliance. The title emphasizes the presence of 2'-FL-HMO, but in the abstract it is not explicitly mentioned until the keywords. Mention it briefly in the Background or Methods section. Mention improvement in symptoms or parental satisfaction related to tolerance and compliance. Consider slightly expanding the summary to provide more detail on tolerance and compliance results. Rephrase the conclusion to emphasize both aspects (e.g., “HRF with 2'-FL HMO was safe, well tolerated, and supported weight gain in infants with suspected CMA or feeding intolerance”).

Comments on the Quality of English Language

Minor editing of English language required

Round 2

Reviewer 1 Report

Comments and Suggestions for Authors

Thank you for the revised manuscript. It is much improved and is clearer. Just one minor edit, remove the word 'than' from the second line of the Introduction - it should be "ranges from approximately 1%-7.5% without the preceding 'than'".

Reviewer 2 Report

Comments and Suggestions for Authors

Accept in present form

Reviewer 3 Report

Comments and Suggestions for Authors

All my suggestions were well taken into account by the authors. Therefore, this manuscript may be considered for publication in this journal.